# Comparative efficacy of chemical and botanical pediculicides in Thailand and 4% dimeticone against head louse, *Pediculus humanus capitis*

Manachai Yingklang[1], Chadaporn Nuchjangreed Gordon[2], Patchana Hengboriboonpong Jaidee[1], Phonpilas Thongpon[3], Somchai Pinlaor[3]*

1 Faculty of Public Health, Burapha University, Chonburi, Thailand, 2 Department of Medical Sciences, Faculty of Allied Health Sciences, Burapha University, Chonburi, Chonburi Province, Thailand, 3 Department of Parasitology, Faculty of Medicine, Khon Kaen University, Khon Kaen, Thailand

* psomec@kku.ac.th

**Data Availability Statement:** All relevant data are within the paper and its Supporting information files.

## Abstract

Head louse infestations remain a global public-health concern due to increased resistance of lice to artificial pediculicides. In Thailand, there is a lack of comparative data on the current efficacy of pediculicides for treating head lice. In this study, we explored the status of botanical and toxic synthetic pediculicides with that of 4% dimeticone liquid gel for treating head lice in Thailand. The *ex-vivo* pediculicidal activity of various pediculicidal shampoos available at drugstores in Thailand was assessed and compared with that of 4% dimeticone liquid gel. The shampoos chosen were based on active ingredients toxic to lice (1% permethrin, 0.6% carbaryl, 0.15% *Stemona* root crude extract, or mixed plant extracts), whereas dimeticone acts physically on lice. We found that exposure to 4% dimeticone liquid gel following the manufacturer's instructions completely killed 100% of head lice in 15 min, whereas other pediculicide products failed to kill the great majority of head lice, whether treatment was for 10 min (resulting in 0% to 50.0% mortality) or 30 min (resulting in 17.0% to 60.0% mortality). We also extended a clinical assessment to confirm the efficacy of 1% permethrin for treating head lice in infested schoolchildren. In this clinical assessment, none of the 26 children treated with 1% permethrin shampoo achieved a cure after two applications. These results highlight that 4% dimeticone demonstrated a higher *ex-vivo* pediculicidal efficacy compared to both chemical and botanical pediculicides in Thailand. Conversely, 1% permethrin showed low efficacy in both laboratory and clinical assessments. Given its physical mode of action, 4% dimeticone merits consideration as an alternative treatment option for lice in Thailand, particularly in cases where treatment with toxic pediculicides has proven ineffective.

## Introduction

The human louse (*Pediculus humanus*) is an ectoparasite that feeds on blood and includes two medically important ecotypes: *Pediculus humanus humanus* Linnaeus, the body louse and

**Funding:** This study was granted by Faculty of Medicine, Khon Kaen University, Thailand to SP. (Grant number RU65201). The funders had no role in study design, data collection and analysis, decision to publish, or preparation of the manuscript.

**Competing interests:** The authors declare that there is no conflict of interest. Any mention of the trademark in this document is purely informational and is not meant to be used for profit or to infringe upon the legal rights of the trademark's owner. This article is simply distributed for scientific purpose.

*Pediculus humanus capitis* De Geer, the head louse [1]. Although these lice share physical similarities, they differ ecologically. The head louse is limited to the scalp and feeds more frequently while the body louse dwells in clothes and feeds on human blood about five times a day [2]. Understanding of genetic diversity and its distribution in lice is largely now based on studies of molecular sequences [3–5]. The human louse found in Thailand is the head louse: *Pediculus humanus capitis* [6, 7], while body louse has never been reported.

Infestation with head lice is generally found in schoolchildren, especially those 3–12 years of age [8, 9]. The prevalence of head louse infestation varies from 15.1% to 86.1% in Thailand [10, 11]. Direct head-to-head contact with infested people is the main means of spread. Transmission is also possible via shared personal items (hairbrushes, clothing, hats, towels, and combs) [8]. Pruritus, induced by saliva of lice, is the most common symptom and scaling of the scalp is a severe consequence associated with chronic lesions. Unlike the body louse, dangerous pathogens (especially *Rickettsia prowazekii*, *Bartonella quintana*, and *Borrelia recurrentis*) can be found in head lice but it is unclear whether the lice can transmit these to humans [12].

Several methods are widely used to control head louse infestations. These include neurotoxic chemical pediculicides (e.g., permethrin, malathion, benzene hexachloride, carbaryl, malathion, and ivermectin), occlusive agents (e.g., benzyl alcohol, isopropyl myristate and dimeticone), and manual removal methods (louse-combing) [13, 14]. Toxic chemical pediculicides are the mainstay of therapy but can have various adverse effects on humans and often fail to kill all lice, especially the egg stage. Head lice have also developed resistance to such pediculicides in many countries [15–18]. To combat resistant lice, it is essential to explore treatment options utilizing medications from different classes [19]. One potential alternative is the use of physically acting agents, with dimeticone being a prominent option, particularly in regions with a high resistance rate [20]. It has been suggested that dimeticone is superior to or equally effective as neurotoxic pediculicides [21].

Dimeticone is a silicone-based polymer or polydimethylsiloxane that kills lice by blocking physical exchange of fluid and gasses between the louse and the environment [22]. It has been widely used to control head lice in European countries and has been shown, both experimentally and clinically, to be highly effective [23–28]. Those studies include 4–100% concentrations of dimethicone based on different modes of use such as shampoo, lotion, and liquid gel.

In Thailand, there have been no reports on the efficacy of dimeticone for treatment of head lice. Neurotoxic chemical pediculicides (permethrin shampoo (0.75% and 1% w/w), carbaryl shampoo (0.6% w/w), or 25% benzyl benzoate lotion) are available in pharmacies. Chemical pediculicides, especially 1% permethrin and 0.6% carbaryl, are used most frequently. Permethrin is a member of the pyrethroid family that disrupts the sodium-channel function of insects, and carbaryl belongs to a group of carbamate insecticides that inhibit acetylcholinesterase, leading to spastic paralysis and death [29]. Pediculicidal products containing medicinal plants (*Stemona* root crude extract, and/or *Annona squamosa* mixed plant extract) are also available in pharmacies and local markets: their mechanism of action against head lice is not understood.

Resistance of lice to pyrethroids has also been reported in Thailand [30] but information about the extent of this resistance needs to be updated. This is necessary to ensure the sustained effectiveness of those pediculicide treatments that are successful in controlling infestation [31]. Utilizing *ex-vivo* studies is an initial step in determining the necessary conditions to kill lice, enabling the assessment of efficacy and prediction of treatment failure or resistance before conducting any clinical trials.

Here, we conducted an *ex-vivo* bioassay aimed to investigate the pediculicidal efficacy of neurotoxic chemical pediculicides (1% permethrin and 0.6% carbaryl) and botanical

pediculicide shampoos (containing 0.15% *Stemona* root crude extract, or mixed plant extracts) available in drugstores in Thailand compared with the physically acting 4% dimeticone liquid gel (Hedrin) (not commercially available in Thailand). The pediculicidal exposure times we used to follow the manufacturer's instructions for each product and was increased to 30 min in one set of experiments. We also extended a clinical investigation to confirm the efficacy of 1% permethrin shampoo before and after treating head lice infestation in schoolchildren.

## Materials and methods

### Human ethical statement

This protocol was approved by the human ethics committees from Khon Kaen University, Thailand (HE621476) in accordance with the Declaration of Helsinki and the ICH Good Clinical Practice Guidelines. Participants were recruited to the study between July and October 2022. Before head louse sample collection and pediculicidal treatment, signed, written and informed consent forms were provided by parents/guardians of schoolchildren after verbal description of the objectives and the method of the research. Participation was entirely voluntary. No photographs or identifying information of the participants are found in the paper. The data were analyzed anonymously. At the end of this study, children who still had head lice infestations were treated with appropriated products (permethrin and/or carbaryl).

### Pediculicides used and head louse sample collection

Four commercial pediculicide products, each with different bioactive components, were purchased from a drugstore in Chonburi Province, Thailand. Dimeticone 4% liquid gel (Hedrin Once) was purchased from the Thornton & Ross Ltd, Huddersfield, UK. The bioactive ingredients, mode of use, and recommended lengths of application for each product are shown in Table 1.

In total, 109 girl schoolchildren (7–12 years of age) from one public primary school in an urban area in Chonburi Province were examined for head lice infestation by trained staff using a fine-tooth comb (with teeth spaced 0.3 mm apart and 10.7 mm in length) and a magnifying glass. Twenty-six (26/109, 23.85%) individuals were found to be infested (presence of motile lice and/or viable eggs). This school had not previously taken part in any head louse prevention projects. Children also had not received any pediculicide treatment at home in the past three months. To collect head lice, a fine-tooth comb was passed through the hair of the volunteers from the scalp to the ends of the hair. Live lice were combed onto a sheet of white paper and transferred to a small plastic box. All head louse samples were kept in plastic containers and

**Table 1. Pediculicides used in this study.**

| Commercial products | Active ingredients | Mode of use | Recommended length of exposure |
|---|---|---|---|
| A | 1% permethrin[a] | Shampoo | 10 min |
| B | 0.6% carbaryl[a] | Shampoo | 5–10 min |
| C | Mixed plant extracts[a] (*Azadirachta indica* Leaf extract, *Sapindus rarak* fruit extract, and *Annona squamosa* extract)[a] | Shampoo | 10 min |
| D | 0.15% *Stemona* root crude extract[a] | Shampoo | 10 min |
| E | 4% dimeticone (Hedrin)[b] | Liquid gel | 15 min |

[a] Commercial product available in drugstores in Thailand

[b] Product not commercially available in Thailand

transported within 30 min to the laboratory, Faculty of Public Health, Burapha University, for experimental testing. The total time elapsing between head lice sample collection and laboratory work was 2 hr.

## Pediculicidal bioassay

A topical bioassay was used to evaluate the pediculicidal efficacy of each head louse treatment. Before testing, each louse was examined for morphological integrity (complete organs and active movement) under a light stereomicroscope. Ten head lice (adults), both males and females, were used in each experimental treatment. A total of 2 ml of each pediculicide tested was spread over the body of lice on filter paper (Whatman No. 1; 9-cm diameter). Batches of head lice were exposed to each solution for 10 or 15 min (as recommended by the manufacturer) or 30 min under 29–30 ˚C; 69% relative humidity. Distilled water was used as a negative control and 1% permethrin shampoo was used as a reference control. At the end of the exposure time, the lice were transferred to another petri dish (90 × 15 mm) and rinsed with distilled water for 2–3 min then transferred to dry filter paper. In the group treated with 4% dimeticone, the lice were rinsed with diluted normal shampoo (not a pediculicide) before washing with distilled water for 2–3 min. Following that, treated lice were observed under a light stereomicroscope and mortality was calculated at intervals up to 8 hr after the end of the exposure period. This observation time was based on the findings of previous workers [24, 32, 33]. The criteria of mortality or no vital sign included that they did not show movement of any appendage (antennae and legs) or internal organ (gut) at the end of the observation time [24, 34]. All experiments were done in triplicate.

## Treatment of head louse infestation in schoolchildren

To explore the efficacy of 1% permethrin shampoo to kill head lice in a clinical assessment, the 26 infested individuals were re-examined two weeks after the initial head louse collection. The severity of the infestation was determined using a fine-tooth comb (for crawling stage) and a magnifying glass (for egg stage) and scored as mild (one to five lice found after 5–6 strokes of the comb or one to five live lice eggs), moderate (six to ten lice after 5–6 strokes) or severe (more than ten lice after 5–6 strokes of the comb or more than ten live lice eggs) [35]. Infested individuals with hair length reaching close to the shoulders were then treated with 15 ml of 1% permethrin shampoo. Those with shorter hair were treated with 10 ml of assigned shampoo. Following administration of the shampoo by trained staff, all treated individuals were required to wear a shower cap for 10 min. In this study, 1% permethrin shampoo was chosen because it is generally thought to be safer than other chemical pediculicides. It is also registered as a pediculicidal drug for control of head lice under the food and drug administration of Thailand. At the end of the application time, the heads of treated individuals were washed with tap water without being combed. Any common side effects (irritation, burning skin, swelling, and red eyes) resulting from the pediculicide product use were immediately evaluated by a nurse. Treatment of head lice was done in two applications; the first treatment was applied on day 0 and the second on day 7. After the first round of treatment, head louse infestation was assessed on day 1 (comb 5–6 strokes) and on day 7. The second treatment on day 7 was immediately after the assessment of persisting infestation. The percentage of cure was calculated by checking the hair and scalp using a fine-tooth comb one week later (day 14) and continuing to the next week (day 21) to confirm the efficacy of lice clearance and no hatched lice. Treatment failure was defined when crawling lice were found on day 1 or day 7 after the first-round treatment and also found on day 14 after the second-round treatment [36].

## Statistical analysis

The mortality of head lice after exposure to neurotoxic chemical pediculicides (1% permethrin and 0.6% carbaryl), botanical pediculicide shampoos (containing 0.15% *Stemona* root crude extract, or mixed plant extracts), and 4% dimeticone liquid gel in *ex-vivo* testing and the cure rates from the clinical intervention were explored using descriptive statistics (percentage, mean ±SE). Relative efficacy of the experimental treatments was analyzed using Kruskal–Wallis and Dunn's tests as implemented in the STATA package version 10.1 (StataCorp LLC, College Station, Texas, United States). Any p-value less than 0.05 was accepted as statistically significant.

## Results

### *Ex-vivo* pediculicidal testing

In the 4% dimeticone liquid gel treatment group when using the same application time with shampoo products in Thailand, 96.7% of lice were dead following a 10-min application. Mortality increased to 100% after exposure for 15 min (the manufacturer's instructions time of dimeticone product) (Fig 1 & S1 File). This was statistically significantly different from the positive control group (1% permethrin) (p < 0.05), in which no lice died. In the group treated with 0.6% carbaryl shampoo, 50% of lice were dead following the recommended application time of 10 min. In addition to 1% permethrin, low efficacy was observed for 0.15% *Stemona* root crude extract and mixed plant extract shampoos, as well as for distilled water (negative control). Efficacy of these treatments remained low at both 30 min and 8 hr after a 30 min exposure: for 0.6% carbaryl, 1% permethrin, 0.15% *Stemona* root crude extract, and mixed plant extract shampoo treatments, efficacy was 60.0%, 43.3%, 20.0%, and 17.0%, respectively (Fig 2 & S1 File).

### Treatment of head louse infestation in schoolchildren

Twenty-six schoolchildren (26/109, 23.85%) were infested with head lice and received two applications of 1% permethrin-based shampoo (on day 0 and day 7). Four of the children had mild infestations, thirteen had moderate infestations and nine had severe infestations. Head lice (adult and/or nymph) were still present on the first and seven days after the first round of treatment, indicating a low efficacy to kill at the crawling lice stage. On assessment days 14 and 21, all participants were still infested with head lice (presence of motile lice), despite a second treatment on day 7 (S2 File). There were no common side effects (irritation, burning skin, swelling, and red eyes) observed that could be associated with the use of 1% permethrin shampoo.

## Discussion

In this study, we explored the current efficacy of chemical and botanical pediculicide products in Thailand compared with 4% dimeticone liquid gel for treating head lice using an *ex-vivo* bioassay. After the recommended application time, dimeticone 4% killed all head lice (100% mortality). Treatment of head lice with 0.6% carbaryl shampoo for 10 min killed only 50.0% of them, while other pediculicide products (1% permethrin, 0.15% *Stemona* root crude extract, and mixed plant extract shampoos) failed to kill most head lice despite their correct use. Almost all lice used in the experimental treatments (other than the dimeticone treatment) recovered after treatment and were still moving and walking 8 hr later, indicating low efficacy and/or resistance to the tested products.

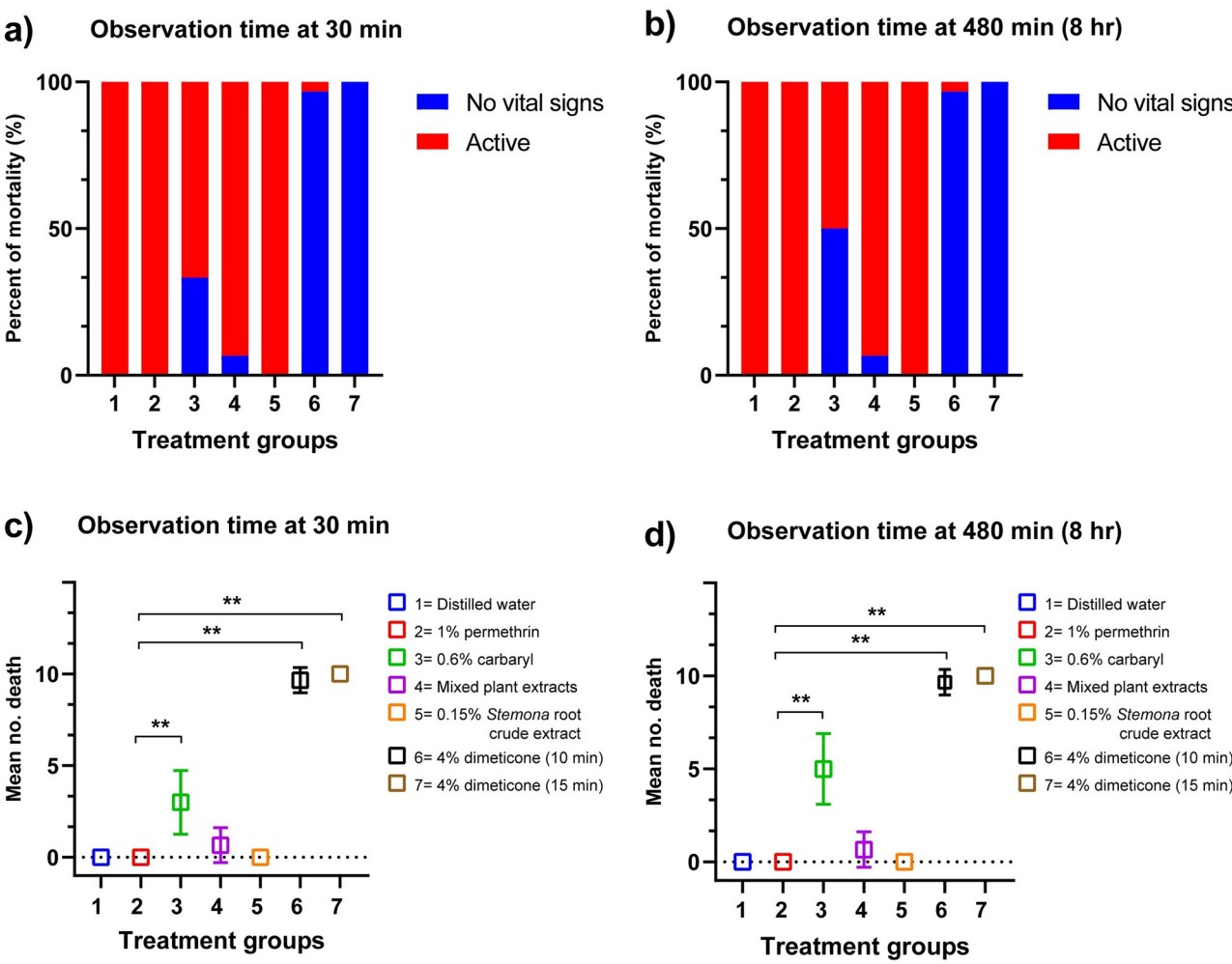

**Fig 1. Percentage mortality of head lice and mean number of deaths recorded at 30 min and 8 hr after the end of a 10 or 15 min exposure to each solution based on the topical test.** a) Percentage mortality of head lice recorded at 30 min and (b) 8 hr after the end of an exposure. c) Mean number of deaths recorded at 30 min and (d) 8 hr after the end of an exposure to each solution. Group 1 = exposure to distilled water; 2 = 1% permethrin; 3 = 0.6% carbaryl; 4 = Mixed plant extracts; 5 = 0.15% *Stemona* root crude extract; 6 = 4% dimeticone (10 min); 7 = 4% dimeticone (15 min). NS = not significant; * = p <0.01; ** = p <0.001 based on Kruskal–Wallis and Dunn's test (compared mean number of lice with reference control group: 1% permethrin). The experiments were conducted in triplicate (n = 10 each group).

It is generally assumed that an artificial pediculicide delivered in a shampoo can be used for a short time on wet hair (5–10 min) to kill crawling louse stages [37]. However, our results indicated that treating head lice with shampoos for the recommended length of time failed to kill most of them. Interestingly, when exposure time was increased to 30 min, the mortality of head lice also increased, but did not result in 100% mortality in any treatment. A long treatment period carries the risk of adverse effects. Unlike compounds with low neurotoxicity (e.g., dimeticone), carbaryl is a synthetic carbamate insecticide [29]. This chemical can have adverse effects on human skin [38]. For long-term use in schoolchildren, the safety and efficacy of this pediculicide should be considered.

In this study, we also extended the intervention pre-post assessments to confirm the efficacy of 1% permethrin for treating head lice infesting schoolchildren. This clinical assessment was done to avoid the limitations of the *ex-vivo* study. To be effective in the control of head lice,

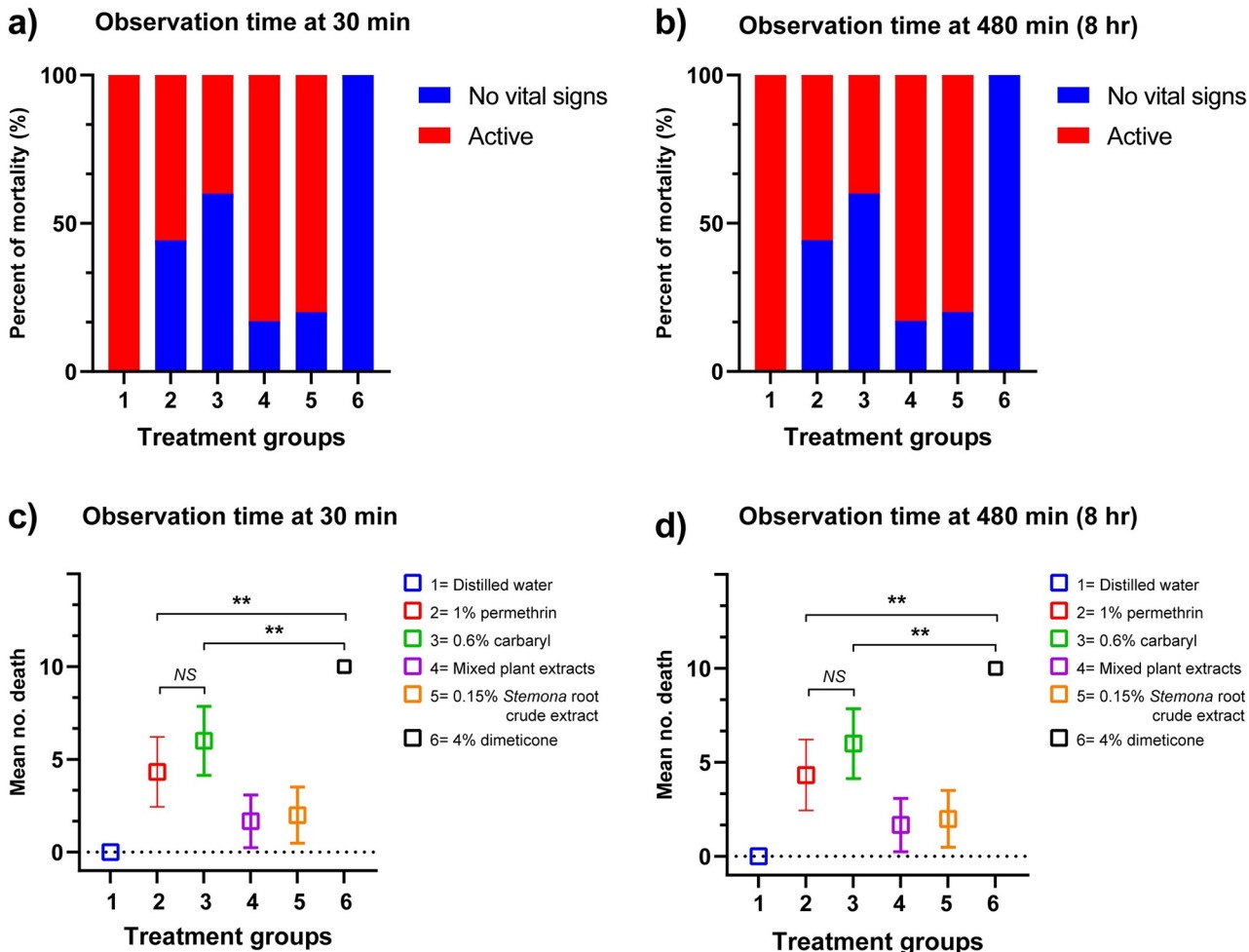

**Fig 2. Percentage mortality of head lice and mean number of deaths recorded 30 min and 8 hr after the end of a 30 min exposure to each solution based on the topical test.** a) Percentage mortality of head lice recorded at 30 min and (b) 8 hr after the end of an exposure. c) Mean number of deaths recorded at 30 min and (d) 8 hr after the end of an exposure to each solution. Group 1 = exposure to distilled water; 2 = 1% permethrin; 3 = 0.6% carbaryl; 4 = Mixed plant extracts; 5 = 0.15% *Stemona* root crude extract; 6 = 4% dimeticone. NS = not significant; * = p <0.01; ** = p <0.001 based on Kruskal–Wallis and Dunn's test (compared mean number of lice with reference control group: 1% permethrin). The experiments were conducted in triplicate (n = 10 each group).

two treatments with pediculicides are needed, a week apart [14]. Our results indicated that 26 infested schoolchildren still had living adults and nymphs on their scalps after both the first and second applications of permethrin. This result was supported by the *ex-vivo* bioassay; 1% permethrin failed to kill any head lice after 10 and 30 min exposures. If some crawling lice survive treatment, the infestation will become re-established [19]. In this study, we did not find any side effects (skin irritation, burning skin, swelling, and red eyes) that could be associated with the use of 1% permethrin shampoo. This could be due to the administration of the shampoo by trained, knowledgeable personnel. However, our study cannot guarantee that using 1% permethrin pesticide is safe because of the small number of children tested. Side effects from pediculicide may be discovered with a larger number of tested. We recommend that children's head lice be treated by their parents, who should be advised to follow the manufacturer's instructions carefully. It is also advisable to seek medical advice before using any treatment

options, especially for children under two years of age or those with allergies or skin sensitivities.

A limitation of this study is the small sample size of head lice used for the experimental study. In Thailand, several cosmetic or homemade shampoo products containing different local plants or herbs are also claimed to be useful for treatment of pediculosis and are available as over-the-counter preparations. We did not test all of these because of the advised requirement for daily administration. Nor did we test the topical 25% benzyl benzoate lotion in the *ex-vivo* bioassay because of its different mode of use and the requirement for an exposure time of 12 to 24 hr. In the clinical assessment, we did not examine head lice in boys, despite their typically short hair, often in the form of an army-style crew cut, and the prevalence of infestation in Thailand has been reported with zero or very low, depending on the specific residential area [10, 11]. However, boys can harbor head lice and transmit them to their classmates. Furthermore, our study did not include family members of infested children or children from different schools within the same community, thereby limiting our examination of head louse infestations. It is important to consider that these factors may potentially contribute to the risk of reinfestation, which cannot be solely attributed to permethrin resistance. Future studies should control these potential risks of reinfestation. In the future, larger samples of head lice should be used, and the efficacy of several additional pediculicide products, particularly those containing herbal components and with different modes of use, should be assessed for the treatment of head louse infestations in schoolchildren.

In this study, we emphasize that the pediculicides commonly sold and authorized by health authorities have demonstrated ineffectiveness in efficiently eliminating lice. This finding helps explain the low *ex-vivo* efficacy and the high rate of infestation among schoolchildren, which has reached 23.85%. In the meantime, use of 4% dimeticone liquid gel is to be encouraged in Thailand. This agent is not neurotoxic and its physical mode of action on head lice means that it can be effective against head lice whether they are resistant or susceptible to other preparations.

## Conclusion

The present study provides evidence that the chemical and botanical commercial pediculicides commonly used in Thailand, including 1% permethrin, 0.6% carbaryl, 0.15% *Stemona* root crude extract, and mixed plant extracts, do not effectively eliminate all lice. This lack of efficacy can be attributed to their low *ex-vivo* efficacy. Specifically, permethrin 1% exhibited limited pediculicidal efficacy in both laboratory and clinical assessments. Dimeticone 4% liquid gel was more effective at killing head lice than other products when used according to the manufacturer's instructions. Therefore, it is advisable for health authorities to promptly introduce dimethicone-based products in the country.

## Supporting information

**S1 File. Percentage mortality of head lice and mean number of deaths recorded at 30 min and 8 hr after the end of an exposure to each solution based on the topical test.**
(DOCX)

**S2 File. Efficacy and safety of 1% permethrin shampoo for treating head lice infestation in children.**
(XLSX)

## Acknowledgments

The authors would like to thank all students who voluntarily participated in school. We would like to acknowledge Prof. David Blair from Publication Clinic KKU, Thailand, for his comments and editing the manuscript.

## Author Contributions

**Conceptualization:** Manachai Yingklang, Somchai Pinlaor.

**Data curation:** Manachai Yingklang, Somchai Pinlaor.

**Formal analysis:** Manachai Yingklang.

**Funding acquisition:** Manachai Yingklang, Somchai Pinlaor.

**Methodology:** Manachai Yingklang, Chadaporn Nuchjangreed Gordon, Patchana Hengboriboonpong Jaidee, Phonpilas Thongpon, Somchai Pinlaor.

**Project administration:** Somchai Pinlaor.

**Supervision:** Somchai Pinlaor.

**Validation:** Manachai Yingklang, Chadaporn Nuchjangreed Gordon, Patchana Hengboriboonpong Jaidee, Phonpilas Thongpon, Somchai Pinlaor.

**Writing – original draft:** Manachai Yingklang.

**Writing – review & editing:** Manachai Yingklang, Chadaporn Nuchjangreed Gordon, Patchana Hengboriboonpong Jaidee, Phonpilas Thongpon, Somchai Pinlaor.

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
