## [Decision Letter · Decision Letter 0]

18 May 2023

PONE-D-23-11571Comparative efficacy of chemical and botanical pediculicides in Thailand and 4% dimeticone against head louse, Pediculus humanus capitisPLOS ONE

Dear Prof.  Pinlaor

Thank you for submitting your manuscript to PLOS ONE. After careful consideration, we feel that it has merit but does not fully meet PLOS ONE’s publication criteria as it currently stands. Therefore, we invite you to submit a revised version of the manuscript that addresses the points raised during the review process.

Please submit your revised manuscript by 02/07/2023. If you will need more time than this to complete your revisions, please reply to this message or contact the journal office at plosone@plos.org. Please include the following items when submitting your revised manuscript:A rebuttal letter that responds to each point raised by the academic editor and reviewer(s). You should upload this letter as a separate file labeled 'Response to Reviewers'.A marked-up copy of your manuscript that highlights changes made to the original version. You should upload this as a separate file labeled 'Revised Manuscript with Track Changes'.An unmarked version of your revised paper without tracked changes. You should upload this as a separate file labeled 'Manuscript'.If applicable, we recommend that you deposit your laboratory protocols in protocols.io to enhance the reproducibility of your results. Protocols.io assigns your protocol its own identifier (DOI) so that it can be cited independently in the future. For instructions see: https://journals.plos.org/plosone/s/submission-guidelines#loc-laboratory-protocols. Additionally, PLOS ONE offers an option for publishing peer-reviewed Lab Protocol articles, which describe protocols hosted on protocols.io. Read more information on sharing protocols at https://plos.org/protocols?utm_medium=editorial-email&utm_source=authorletters&utm_campaign=protocols.

We look forward to receiving your revised manuscript.

Kind regards,

Joshua Kamani, PhD

Academic Editor

PLOS ONE

Journal Requirements:

Additional Editor Comments (if provided):

Dear Pinlaor

I'd like to make a suggestion for the modification of the topic to- Comparative efficacy of chemical and botanical pediculicides and 4% dimeticone against head louse, Pediculus humanus capitis in Thailand

Reviewers' comments:

Reviewer's Responses to Questions

**Comments to the Author**

1. Is the manuscript technically sound, and do the data support the conclusions?

Reviewer #1: Yes

Reviewer #2: Partly

2. Has the statistical analysis been performed appropriately and rigorously? 

Reviewer #1: I Don't Know

Reviewer #2: Yes

3. Have the authors made all data underlying the findings in their manuscript fully available?

Reviewer #1: No

Reviewer #2: Yes

4. Is the manuscript presented in an intelligible fashion and written in standard English?

Reviewer #1: No

Reviewer #2: Yes

5. Review Comments to the Author

Reviewer #1: Lines 32-33: It might be better to give a range of percentage mortality

Lines 35-36: Are you saying that none of the 26 children treated with 1% permethrin was cured after two treatments?

Lines 50-51: It does not appear in this publication that body louse feeds once or twice a day!

Lines 52-53: Does it mean that body lice were never found in Thailand?

Lines 69-70: in the reference number 20 it is rather said that…. : In treating head louse infestation, evidence suggests occlusive agents may be superior to or equally efficacious as neurotoxic pediculicides.

Line 114: Pediculicides used might be better than “Chemical preparation”

Line 121: Was only one kind of louse comb used and if yes if you could give any details about the comb

Lines 254-: It is suggested that the conclusion should be stronger than written here and should include that the most sold and authorized by the health authorities pediculicides are not effective, which would also explain the high rate of infestation of children in the country and it would be advisable to the health authorities to introduce dimethicone-based products such as Hedrin ASAP to the country.

Reviewer #2: The manuscript presents the outcomes of an important study in Thailand, where head lice infestation (HLI) is a significant problem among primary school children, as in many parts of the world. The study is mostly well-designed and well-written and its outcomes are significant as they include the first assessment results of dimeticone in Thailand, presents ex vivo results of local products used in HLI, together with efficacy of permethrin 1% among school children. The content of the study is very rich and there is a hard work in the background of the study. However, an important mistake seems to be effective in the results of HLI-positive school children investigation with 1% permethrin. The study universe had 109 primary school girls only, not any boys. In our experience, despite their short hair compared to girls, boys can also harbor head lice and transmit them to their classmates. As the boys were not investigated nor treated with 1%permethrin on Day 1 and 7, the HLI-positive girls identified on days 7 and 14 may be due to reinfestation from their classmate boys, not because of permethrin resistance. Since permethrin resistance have been reported from many regions in the world, it is not surprising to identify such a resistance also in Thailand, but the design of the study does not exclude the risk of reinfestation from classmates. This potential risk should be mentioned within the drawbacks of the study.

6. PLOS authors have the option to publish the peer review history of their article (what does this mean?). If published, this will include your full peer review and any attached files.

Reviewer #1: No

Reviewer #2: No

---

## [Author Response · Author response to Decision Letter 0]

2 Jun 2023

We are grateful for the opportunity provided by the editor and reviewers to revise the manuscript. We sincerely appreciate the reviewers' insightful criticism and valuable recommendations, which have greatly contributed to improving the quality of the manuscript. All of the reviewers' suggestions and comments have been thoroughly addressed.

Reviewer #1: 

Lines 32-33: It might be better to give a range of percentage mortality

Response: We have provided a range of percentage mortality as “…..whether treatment was for 10 min (resulting in 0 to 50.0% mortality) or 30 min (resulting in 17.0 to 60.0% mortality)……..”

Lines 35-36: Are you saying that none of the 26 children treated with 1% permethrin was cured after two treatments?

Response: Yes, we have modified the sentence as “none of the 26 children treated with 1% permethrin shampoo achieved a cure after two applications.”

Lines 50-51: It does not appear in this publication that body louse feeds once or twice a day!

Response: We acknowledge this comment and have modified to “body louse feeds about five times a day”. 

Lines 52-53: Does it mean that body lice were never found in Thailand?

Response: Yes, there aren’t any previous reports of body lice being found in Thailand. We have modified the sentence as “ The human louse found in Thailand is the head louse: Pediculus humanus capitis [6, 7], while body louse has never been reported.”

Lines 69-70: in the reference number 20 it is rather said that…. : In treating head louse infestation, evidence suggests occlusive agents may be superior to or equally efficacious as neurotoxic pediculicides.

Response: We have modified accordingly and also included additional references.

“To combat resistant lice, it is essential to explore treatment options utilizing medications from different classes [20]. One potential alternative is the use of physically acting agents, with dimeticone being a prominent option, particularly in regions with a high resistance rate [21]. It has been suggested that dimeticone is superior to or equally effective as neurotoxic pediculicides [22].”

Line 114: Pediculicides used might be better than “Chemical preparation”

Response: We have changed “Chemical preparation” to “Pediculicides used” accordingly.

Line 121: Was only one kind of louse comb used and if yes if you could give any details about the comb

Response: We have added more details of the louse comb as “….using a fine-tooth comb (with teeth spaced 0.3 mm apart and 10.7 mm in length)….”

Lines 254-: It is suggested that the conclusion should be stronger than written here and should include that the most sold and authorized by the health authorities pediculicides are not effective, which would also explain the high rate of infestation of children in the country and it would be advisable to the health authorities to introduce dimethicone-based products such as Hedrin ASAP to the country.

Response: We have revised it according to the suggestions provided by the reviewer in the conclusion section.

Reviewer #2: The manuscript presents the outcomes of an important study in Thailand, where head lice infestation (HLI) is a significant problem among primary school children, as in many parts of the world. The study is mostly well-designed and well-written and its outcomes are significant as they include the first assessment results of dimeticone in Thailand, presents ex vivo results of local products used in HLI, together with efficacy of permethrin 1% among school children. The content of the study is very rich and there is a hard work in the background of the study. However, an important mistake seems to be effective in the results of HLI-positive school children investigation with 1% permethrin. The study universe had 109 primary school girls only, not any boys. In our experience, despite their short hair compared to girls, boys can also harbor head lice and transmit them to their classmates. As the boys were not investigated nor treated with 1% permethrin on Day 1 and 7, the HLI-positive girls identified on days 7 and 14 may be due to reinfestation from their classmate boys, not because of permethrin resistance. Since permethrin resistance have been reported from many regions in the world, it is not surprising to identify such a resistance also in Thailand, but the design of the study does not exclude the risk of reinfestation from classmates. This potential risk should be mentioned within the drawbacks of the study.

Response: We acknowledge your insightful comments on our manuscript. We also appreciate your attention to detail and the valuable points you have raised regarding the potential risk of reinfestation from classmates, particularly boys, in our study on HLI-positive school children treated with 1% permethrin.

We understand your concern that boys, despite having shorter hair compared to girls, can also harbor head lice and transmit them to their classmates. However, it is important to note that previous studies conducted in Thailand have reported a very low or no infestation rate among boys, suggesting that their contribution to the reinfestation dynamics in our study population may be minimal. Nevertheless, we have included the importance of considering potential reinfestation sources, such as classmates in our limitation section to provide a comprehensive analysis of the findings and emphasize the importance of considering this aspect in future investigations as suggestion in the Discussion.

---

## [Decision Letter · Decision Letter 1]

8 Jun 2023

Comparative efficacy of chemical and botanical pediculicides in Thailand and 4% dimeticone against head louse, Pediculus humanus capitis

PONE-D-23-11571R1

Dear Dr. Pinlaor,

We’re pleased to inform you that your manuscript has been judged scientifically suitable for publication and will be formally accepted for publication once it meets all outstanding technical requirements.

Kind regards,

Joshua Kamani, PhD

Academic Editor

PLOS ONE

Additional Editor Comments (optional):

Dear sir

I am glad to inform you that your manuscript has been accepted for publication.

Reviewers' comments:

Reviewer's Responses to Questions

**Comments to the Author**

1. If the authors have adequately addressed your comments raised in a previous round of review and you feel that this manuscript is now acceptable for publication, you may indicate that here to bypass the “Comments to the Author” section, enter your conflict of interest statement in the “Confidential to Editor” section, and submit your "Accept" recommendation.

Reviewer #1: All comments have been addressed

Reviewer #2: All comments have been addressed

2. Is the manuscript technically sound, and do the data support the conclusions?

Reviewer #1: Yes

Reviewer #2: Yes

3. Has the statistical analysis been performed appropriately and rigorously? 

Reviewer #1: Yes

Reviewer #2: Yes

4. Have the authors made all data underlying the findings in their manuscript fully available?

Reviewer #1: Yes

Reviewer #2: Yes

5. Is the manuscript presented in an intelligible fashion and written in standard English?

Reviewer #1: Yes

Reviewer #2: Yes

6. Review Comments to the Author

Reviewer #1: I think that all questions of the reviewers have been answered by the authors and accordingly the manuscript could be published

Reviewer #2: It is clear that the authors have clearly reviewed the comments of the referees, and revised the manuscript according according to their suggestions. In my opinion, the manuscript is complete now and it may be accepted for publication. It will contribute to the research and infection data in Thailand.

7. PLOS authors have the option to publish the peer review history of their article (what does this mean?). If published, this will include your full peer review and any attached files.

Reviewer #1: No

Reviewer #2: **Yes: **Prof. Özgür Kurt, M.D., Ph.D.

---

## [Editor Report · Acceptance letter]

15 Jun 2023

PONE-D-23-11571R1 

Comparative efficacy of chemical and botanical pediculicides in Thailand and 4% dimeticone against head louse, *Pediculus humanus capitis*

Dear Dr. Pinlaor:

I'm pleased to inform you that your manuscript has been deemed suitable for publication in PLOS ONE. Congratulations! Your manuscript is now with our production department. 

Kind regards, 

on behalf of

Dr. Joshua Kamani 

Academic Editor

PLOS ONE